# Quantitative comparative analysis of human erythrocyte surface proteins between individuals from two genetically distinct populations

Benjamin J. Ravenhill[1], Usheer Kanjee[2], Ambroise Ahouidi[3], Luis Nobre[1], James Williamson[1], Jonathan M. Goldberg[2], Robin Antrobus[1], Tandakha Dieye[3], Manoj T. Duraisingh[2] & Michael P. Weekes [1]

Red blood cells (RBCs) play a critical role in oxygen transport, and are the focus of important diseases including malaria and the haemoglobinopathies. Proteins at the RBC surface can determine susceptibility to disease, however previous studies classifying the RBC proteome have not used specific strategies directed at enriching cell surface proteins. Furthermore, there has been no systematic analysis of variation in abundance of RBC surface proteins between genetically disparate human populations. These questions are important to inform not only basic RBC biology but additionally to identify novel candidate receptors for malarial parasites. Here, we use 'plasma membrane profiling' and tandem mass tag-based mass spectrometry to enrich and quantify primary RBC cell surface proteins from two sets of nine donors from the UK or Senegal. We define a RBC surface proteome and identify potential *Plasmodium* receptors based on either diminished protein abundance, or increased variation in RBCs from West African individuals.

[1] Cambridge Institute for Medical Research, University of Cambridge, Hills Road, Cambridge CB2 0XY, UK. [2] Department of Immunology and Infectious Diseases, Harvard T.H. Chan School of Public Health, Boston, MA, USA. [3] Laboratory of Bacteriology and Virology, Le Dantec Hospital, Cheikh Anta Diop University, Dakar, Senegal. Correspondence and requests for materials should be addressed to M.P.W. (email: mpw1001@cam.ac.uk)

Red blood cells (RBCs) are the most abundant circulating human cell type and are essential for blood gas transport. The inter-individual variation in red blood cell surface proteins defines the range of clinically observed blood types, from the main Rhesus grouping to more minor Landsteiner-Weiner system[1]. Appreciation of the variability of RBC surface proteins is vital to the understanding of transfusion compatibility between donors and recipients, and conditions such as haemolytic disease of the newborn[2].

Proteins at the surface of RBC also play a critical role in susceptibility to blood borne diseases. For example, an increasing number of proteins are recognised as receptors for *Plasmodium* species, causative agents of malaria. Antigens of the Duffy blood group system were classically described as receptors for *Plasmodium vivax*[3]. More recently, additional molecules have been identified as receptors for *P. vivax* or *P. falciparum* species. These include basigin, complement receptor 1 (CR1/CD35), CD55 (complement decay-accelerating factor/Cromer blood group), CD44 (Indian blood group), glycophorin molecules, the Langereis blood group protein ABCB6 and Transferrin receptor (TFRC)[4–10]. Long-standing selective pressures from pathogens such as *Plasmodium* species in populations in which malaria is endemic have led to selection against Duffy antigens[11,12]. For example, more than 95% of West African populations are Duffy negative[13]. Individuals who are genetically null for other receptors have also been reported, including CD55 and ABCB6[14,15]. Furthermore, variable expression of CR1 is well recognised, and the level of CR1 expression correlates with binding to the *P. falciparum* ligand PfRh4[5]. It is therefore possible that selective pressures from *Plasmodium* may have driven variation in other cell surface proteins that have as yet unknown roles in parasite invasion, manifest either as a genetic null phenotype, or more subtly as increased variation in the level of surface expression of the molecule in highly exposed compared to unexposed populations.

There have been several previous studies of the red blood proteome, although none directed specifically at proteins exposed at the cell surface. As >98% of the protein content of RBC is composed of haemoglobin subunits and carbonic anhydrase, proteomic measurement of other less abundant proteins has been more complex. Various methods have been used to selectively enrich these proteins for proteomics, ranging from isolation of 'white ghost' membranes, to absorption of the highly abundant haemoglobin prior to analysis[16–25]. These techniques nevertheless still suffer from contamination by abundant intracellular proteins and offer limited insight into which proteins have extracellular domains.

Here we used selective oxidation and aminooxybiotinylation to label and enrich red blood cell surface proteins, ('Plasma Membrane Profiling', PMP)[7,26]. We have previously shown that this approach has the power to quantify the majority of all cell surface proteins, identifying for example a critical drug transporter in cells latently infected with cytomegalovirus, in addition to a novel malaria receptor[6,27]. By combining this technology with multiplexed tandem-mass tag (TMT)-based MS3 proteomics, we were able to measure the variation in abundance of individual cell surface proteins from individual to individual within nine donors from each of two geographically and genetically distinct human populations, one from West Africa (Senegal) and one from the United Kingdom. We reasoned that either decreased protein abundance or increased variation in protein expression from Senegalese compared to UK RBC would enable identification of putative *Plasmodium* receptor candidates for future characterisation. Finally, by comparing our results to previously published accounts of whole or membrane-enriched RBC proteomes, we defined a 'sensitive' and 'stringent' list of RBC surface proteins.

## Results

**Optimised enrichment of RBC for proteomic analysis.** Previous analyses of the RBC proteome have been complicated by the presence of contaminants in the RBC sample, including white blood cells, platelets and serum proteins. A simple method was therefore developed to isolate fresh RBC for proteomic analysis at maximum purity and yield, with minimum ex vivo manipulation.

Methods for enrichment have previously included combinations of strategies to: deplete leukocytes (α-cellulose, density gradient or simple centrifugation and negative selection using magnetic beads), remove platelets (PBS washing, and positive erythrocyte selection using magnetic beads) and remove plasma constituents (simple centrifugation and PBS washing)[16,17,19–23]. A selection of these methods was compared, initially using flow cytometry as a readout to measure residual contamination with platelets and leukocytes (Supplementary Fig. 1a–c). A sensitive proteomic approach then measured contamination by serum components (Supplementary Fig. 1d–g, Supplementary Data 1). An optimised method involved centrifugation of $2 \times 1$ ml anticoagulated blood at 1000 g for 5 min followed by isolation of 150 µl of packed RBC from the very bottom of each Eppendorf tube. Cells were resuspended in PBS, passed through a Plasmodipur filter then washed two further times.

**Proteomic analysis of RBC PM proteins.** Previous studies have attempted to analyse the RBC membrane proteome by cellular lysis followed by isolation of the membrane fraction ('white ghosts')[25,28], or by metal ion chromoaffinity to deplete haemoglobin[16]. However, both are still subject to substantial contamination by abundant soluble intracellular proteins including haemoglobin, and lack specificity for surface proteins with extracellular domains. The use of 'Plasma Membrane Profiling' (PMP) specifically enriches cell surface glycoproteins via selective oxidation then aminooxy-biotinylation with reagents that are membrane impermeable at 4 °C. Isolation of biotinylated proteins using streptavidin beads facilitates removal of major cytosolic contaminants including haemoglobin by extensive washing of beads using stringent conditions. Relative quantitation of each protein across all donors could be achieved after protein digestion with trypsin by labelling peptides with TMT tags followed by MS3 mass spectrometry[10,26] (Fig. 1a).

We applied these techniques to two sets of fresh RBC samples from nine donors with Caucasian UK heritage and nine donors with Senegalese heritage, identifying 1260 proteins with 1041 at the level of two or more peptides. To identify high confidence PM proteins, a stringent approach required ≥2 peptides for each identified protein and at least one of the following annotations derived from the Uniprot subcellular localisation database: 'Multipass', 'GPI anchored', 'Lipid Anchored', 'Type I transmembrane', 'Type II transmembrane', 'Type III transmembrane', 'Type IV transmembrane', or ≥1 predicted transmembrane regions based on TMHMM version 2.0[29] (Fig. 1b, Methods). Six proteins likely to be abundant serum contaminants were excluded (Clusters C1–C2, Supplementary Fig. 1d, Supplementary Data 1). This yielded a list of 241 and 256 cell surface proteins from UK and Senegalese populations respectively, with 267 proteins quantified in either population and 230 proteins quantified in both (Supplementary Data 2a–b).

**A consensus RBC surface proteome.** Previous applications of proteomics to characterise RBC proteins have classified substantially different numbers of proteins at the cell surface. To develop a consensus RBC cell surface proteome, our data was initially compared to 10 previous publications that used high-throughput proteomics to measure the total RBC, or RBC

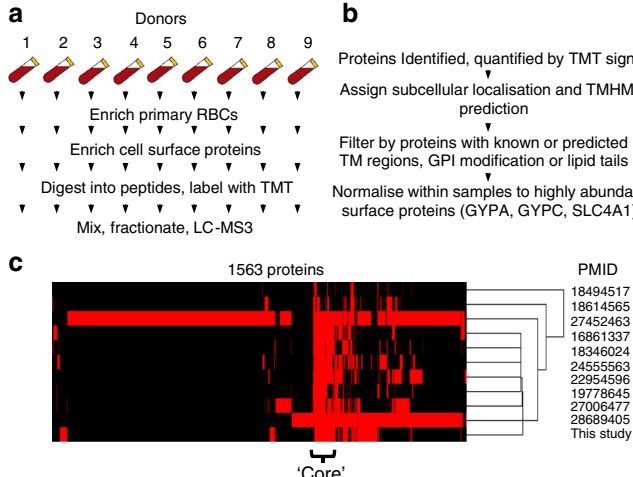

**Fig. 1** Experimental details. **a** Schematic of the experimental workflow. **b** Schematic of workflow for filtering identified proteins. **c** Comparison of this study to ten previous proteomic studies that examined either whole RBC or enriched RBC membranes. Proteins without transmembrane regions or membrane anchors were excluded as described in the text. A binary outcome is shown for 1563 proteins, representing presence (red) or absence (black) of each protein in each study. Data were clustered by uncentered correlation

membrane proteome, and applied the same subcellular location/ TMHMM filters described above (Fig. 1a–c)[16,17,19–25]. A total of 1563 proteins were identified in one or more studies (Fig. 1c, Supplementary Data 3a–b). Hierarchical cluster analysis revealed a central 'core' of high confidence RBC surface proteins, with 160 proteins found in our experiments and at least three previously published studies ('stringent criteria', Supplementary Data 3c), and 240 proteins identified by a minimum of our experiments and at least one previous study ('sensitive criteria', Supplementary Data 3d). These included key blood group proteins GYPA, GYPC, BCAM, SLC4A1, KEL, AQP1, CD55, SLC14A1, XK and RHCE. The utility of these criteria to characterise genuine RBC PM proteins was illustrated by identifications of contaminating platelet markers ITGB3 and ITGA2B, which are not expressed by mature erythrocytes. Both entries were present in the 'sensitive' list, but neither on the 'stringent' list, and were likely to be detected due to the presence of a very low percentage of residual platelets (<0.01%, Supplementary Fig. 1b).

Figure 1b also highlights that a substantial number of proteins were only identified in one or two of the ten previous studies. These include a number of proteins found at high abundance in human serum that are likely to be contaminants, for example complement component C1Q and apolipoproteins L1 and L2 (Supplementary Data 3b). 27 proteins were only identified in the present study (Supplementary Data 3e). Some of these are likely to represent previously undetected low-abundant RBC PM proteins, enriched by our plasma membrane-specific technique, including four multi-pass transmembrane solute carrier (SLC) transporters. At least eight are likely to be contaminants from serum or other low-abundant contaminating cells (Supplementary Data 3e). Despite using a stringent RBC isolation protocol, the presence of remaining low abundance serum contaminants in our study highlight the high sensitivity of the mass spectrometers used. Combining and comparing multiple RBC proteomic datasets from the literature and acquired in orthogonal manners is therefore an important step to define a true RBC surface proteome.

## Composition of the surface proteome.

Intensity based absolute quantitation (iBAQ) was employed to compare the copy number of each protein in the 'sensitive' list, and thus the contribution of each protein to the RBC cell surface proteome[30]. Since iBAQ is based on the summed intensity for each protein (divided by the number of observed peptides), values effectively represent an average across nine donors for each population. This offers the opportunity to provide precise information on the overall population abundance of each PM protein, independent of individual genetic variation. iBAQ values from both populations correlated well ($r^2 = 0.81$), with the best quantified (most abundant) proteins most similarly expressed. The most highly expressed proteins were present at ~$10^5$ greater abundance than the least expressed proteins (Fig. 2a). This dynamic range corresponds well to previous data; for example, in the original description of the iBAQ algorithm, Schwanhäusser et al. found that iBAQ abundances of proteins in NIH3T3 fibroblasts also spanned more than five orders of magnitude[30].

The four most abundant surface proteins, GYPC, GYPA, SLC2A1 and SLC4A1 accounted for ~62–75% of the surface proteome, with GYPC alone representing ~25% (Fig. 2b). Quantified cell surface blood group antigens (which do not include SLC2A1) accounted for ~72–75% of the surface proteome (Supplementary Fig. 2, Supplementary Data 2a, 4). Despite multiple independent analyses by mass spectrometry, no unique peptides corresponding to GYPB were identified. GYPD is an abridged form of GYPC with an alternative start site, and no unique N-terminal GYPD peptide was identified.

A subset of candidate receptors for *Plasmodium* species include proteins whose abundance is lower in Senegalese compared to UK populations. As expected, Duffy antigen was only detected in UK but not Senegalese populations (Supplementary Data 2c). Several proteins also exhibited overall statistically lower abundance in RBC samples from Senegal, for example known *Plasmodium* receptors CD55 (15.5-fold lower) and CD44 (3.1 fold lower) (Fig. 2a, Supplementary Fig. 3, Supplementary Data 2a)[6,7]. Other proteins exhibiting either of these patterns include the solute carrier SLC44A2, the complement regulatory protein CD59 and tetraspanin CD151 (Fig. 2a).

## Variability between genetically distinct populations.

Certain cell surface proteins exhibit a genetically determined high degree of inter-donor variability in expression, including CR1 and complement molecules C4A and C4B[31,32]. By using highly multiplexed TMT-based proteomics, the present study offered the opportunity to directly measure the variability in expression of whole RBC surface proteomes across multiple donors. Despite extensive washing of biotinylated proteins when bound to streptavidin beads, variable levels of contamination with haemoglobin were nevertheless detectable. As opposed to normalisation of data assuming equal protein loading across all channels, normalisation was instead performed from the summed signal: noise values of the three most abundant RBC surface blood group proteins, Glycophorin A (GYPA), Glycophorin C (GYPC) and SLC4A1. All three exhibited low variability across all donors studied from both populations, independent of the method of normalisation used, validating their usefulness for this purpose (Supplementary Fig. 4).

The majority of proteins identified in both populations exhibited relatively little variability (61.8 % of proteins <25% variability in both proteins, Fig. 3a, Supplementary Data 2a, e; Data 2e is an interactive spreadsheet that can be used to visualise any of the data in this manuscript). Using a Mann–Whitney U test we determined that the distribution of %CV values between both populations were similar (p-value for alternative hypothesis of different distributions = 0.103, thus supporting the null hypothesis, established via Monte Carlo method with 100,000

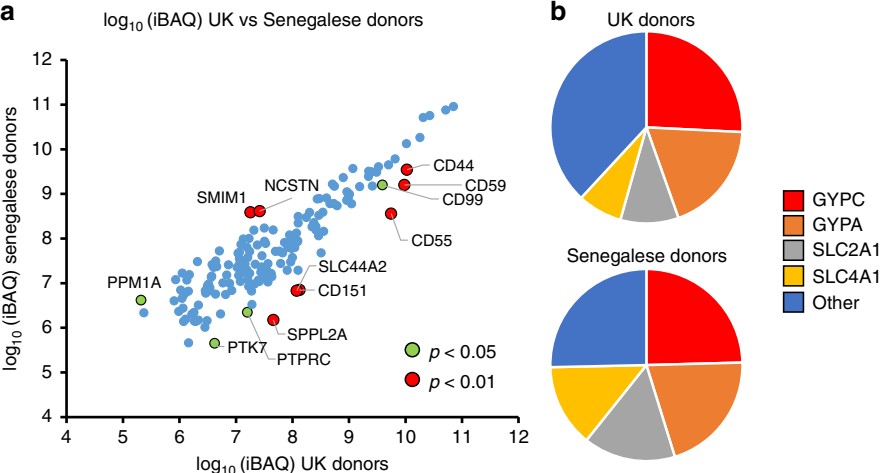

**Fig. 2** Comparison of RBC surface protein abundance between populations. **a** iBAQ abundance values for each protein from all nine UK and all nine Senegalese donors. Data are shown where iBAQ values were available from both populations, for the 'sensitive' list of RBC surface proteins with likely serum contaminants removed. Significance A values were used to estimate $p$-values for ratio of iBAQ abundances for each protein as described in 'Methods'. **b** iBAQ abundances of GYPA, GYPC, SLC2A1 and SLC4A1 as a proportion of the total RBC surface proteome

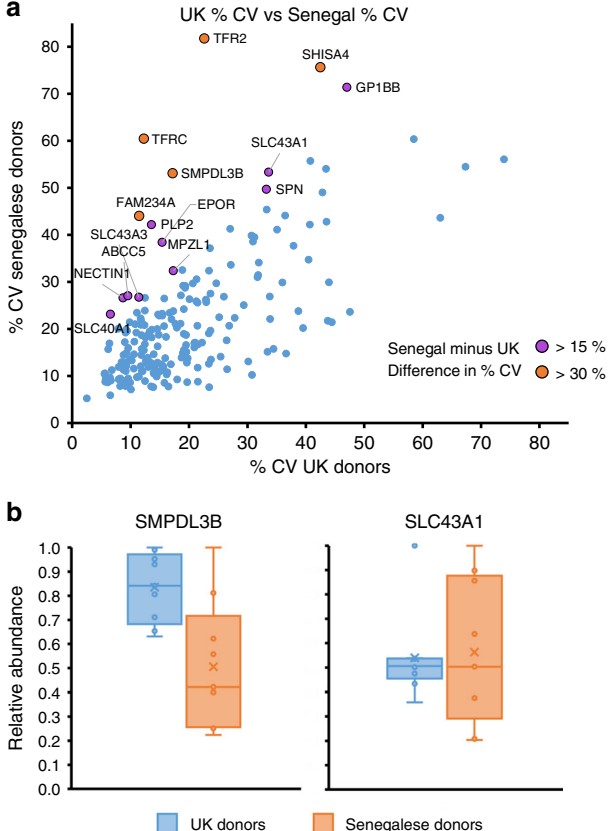

**Fig. 3** Comparison of variability in RBC surface protein expression between populations. **a** Coefficients of variation for each protein identified by 'sensitive' criteria and quantified in both populations. Leukocyte-derived contaminants HLA-A, B and C as well as the X-linked glycoprotein XG exhibited >80% CV and are not shown on this plot to enable easier visualisation of proteins with 0–80% CV. Increased variation was exhibited by some proteins from Senegalese populations, shown in green and red as indicated in the legend. **b** The relative abundance of SMPDL3B and SLC43A1 from each donor in both populations, normalised to a maximum of 1, plotted as a box and whisker plot showing mean, median and interquartile ranges. $n = 9$ biologically independent samples for each group

iterations). A small number of proteins were significantly more variable in the Senegalese compared to the UK population, including Sphingomyelin Phosphodiesterase Acid Like 3b (SMPDL3B) and Solute Carrier 43 Member 1 (SLC43A1). Other candidate molecules with high variability in the Senegalese population included the SLC transporter SLC40A1 (Ferroportin), in which the Q248H mutation has recently been reported to be positively selected to protect against severe malaria[33], and amino acid or nucleobase transporter SCL43A3[34,35]. The poorly characterised protein Family With Sequence Similarity 234 Member A (FAM234A; also known as Integrin Alpha FG-GAP Repeat Containing 3, ITFG3), which is partially deleted in a novel α-thalassemia trait[36] additionally exhibited much higher variability in Senegalese samples.

To determine if measured protein variability was a product of decreased cell surface abundance, the correlation between the percentage coefficient of variance (%CV; standard deviation divided by the mean) and iBAQ values in both populations was examined. Data were poorly correlated, as would be expected from an unbiased assessment of variability. As expected, the highly abundant proteins used for data normalisation including GYPA exhibited some of the lowest variability (Supplementary Fig. 5).

Flow cytometry was used to validate proteomic data. As only UK RBC donors were readily available for confirmatory experiments, a fresh set of UK donor samples was used to examine the variability in two relatively invariant, and two relatively more variable cell surface proteins. The proteins GYPA and SLC4A1 were amongst the least variable proteins in the UK donor RBC samples (Supplementary Fig. 5), which was confirmed by flow cytometry (Fig. 4a, b).

The blood group antigens ICAM4 (Landsteiner-Weiner) and CD35/CR1 (Knops) were variably expressed between donors and of intermediate abundance (Supplementary Fig. 5) suggesting flow cytometry-based assessment would be likely to yield a precise result. The %CV in median fluorescence intensity for both proteins corresponded very well to proteomic data, even though measurements were made on samples from different donors, validating this approach (Fig. 4a, b, Supplementary Data 5). Furthermore, studies of RBC from two donors over four weeks suggested that the cell surface proteins studied are stably expressed over time within an individual (Supplementary Fig. 6).

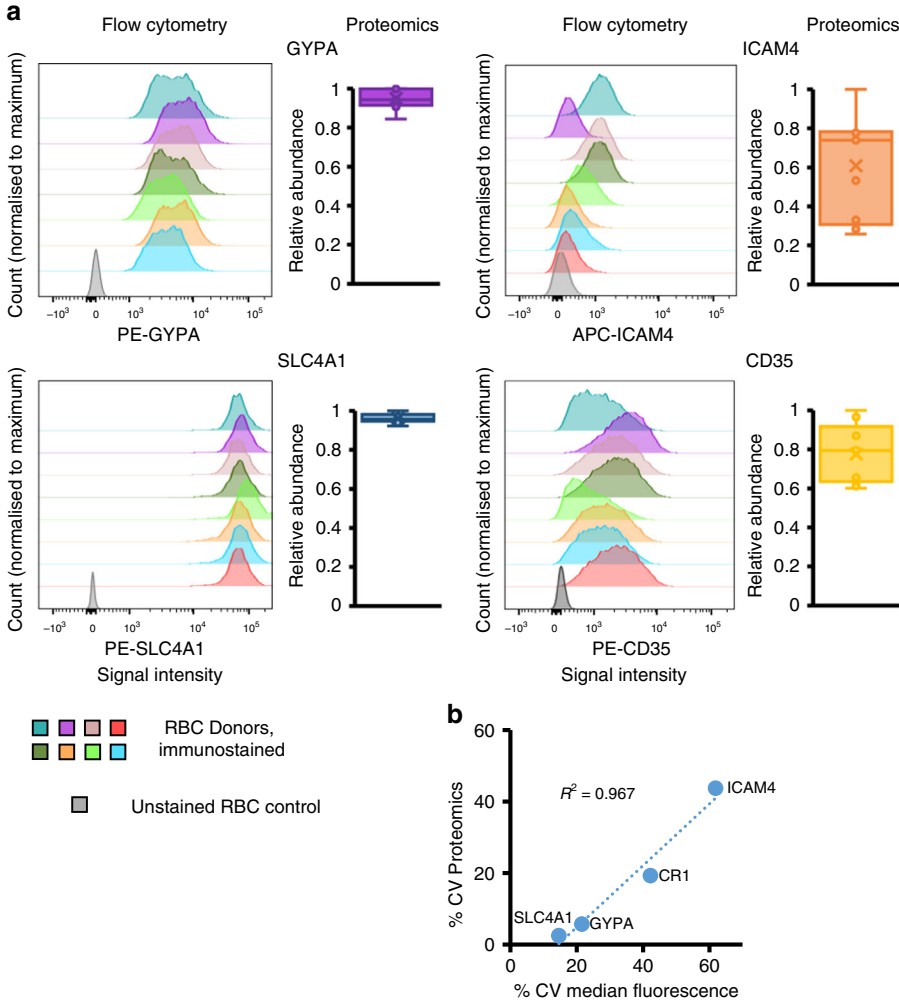

**Fig. 4** Validation of proteomic assessment of expression of cell surface proteins by flow cytometry. **a** Variability of expression of RBC surface proteins from an independent set of UK donors assessed by flow cytometry. Fluorescent signal intensity is shown on the x-axis, and normalised count on the y-axis. Relative abundance of the same four surface proteins as determined by proteomics from the original set of nine UK donors is plotted as an adjacent box and whisker plot showing mean, median and interquartile ranges, normalised to the maximum for each sample. **b** Comparison of %CV values from proteomics with %CV of the median fluorescence intensity from flow cytometry for all four RBC surface proteins (n for flow cytometry = 7 or 8, n for proteomics = 9)

## Discussion

RBC have vital roles in vertebrate physiology. Although a number of studies have examined the composition of the whole, or membrane RBC proteome, none have used an approach specifically directed at the cell surface. An appreciation of which proteins are expressed at the RBC surface, their relative abundance and variation in expression between different populations is vital to understanding diseases caused by pathogens including *Plasmodium* and *Babesia*.

In this study we used a plasma membrane-targeted proteomic approach to compare the composition and variability of the RBC surface proteome between UK and Senegalese populations. Our selective enrichment of cell surface glycoproteins via specific amino-oxybiotinylation prior to RBC lysis offers inherent advantages over previous strategies. In particular, this enables extensive on-bead washing to limit sample contamination with highly abundant intracellular proteins, and selective targeting of molecules with extracellular domains to distinguish surface-expressed proteins from those residing exclusively in intracellular membrane pools. In combination with stringent RBC isolation procedures and bioinformatic analysis, this enabled identification

of 240 high confidence RBC cell surface proteins, and assessment of the relative contribution of each to the RBC surface proteome. Our data will thus form an important resource for future investigations of the red blood cell surface in health and disease.

Phenotypic variation between RBCs from different donors is a critical driver of transfusion reactions, and plays important roles in determining susceptibility to specific pathogens or stresses. Our application of highly multiplexed TMT-based MS3 mass spectrometry enabled an unprecedented comparison of protein expression both between multiple individuals from a given population, and between RBC from distinct populations. Nevertheless, as our studies were necessarily limited to examining nine individuals from each population due to the multiplexing limits of TMT, data may not fully represent variation at a whole population level, and the statistical relevance of these data may similarly be limited. In addition to recapitulating known positive controls (such as detection of Duffy antigen only in UK but not Senegalese samples), a number of proteins exhibited significantly decreased abundance on Senegalese RBCs. These included complement regulatory proteins CD55 (Decay Accelerating Factor) and CD59 (Membrane Attack Complex Inhibition Factor) in

addition to the presumptive choline transporter and thrombosis susceptibility protein SLC44A2[37]. Other amino acid transporters exhibited greater variability in Senegalese compared to UK RBCs, including SLC43A1 and SLC43A3. Although mature red blood cells are not transcriptionally or translationally active, amino acid metabolites are nevertheless used as substrates to synthesise glutathione, which is critical for maintaining the redox state of the red blood cell[38]. Other red blood cell polymorphisms that maintain redox state, for example Glucose-6-Phosphate Dehydrogenase deficiency, have been established to play a protective role against malaria[39]. FAM234A protein was also significantly more variably expressed in Senegalese donors. The function of FAM234A is poorly characterised, although single nucleotide polymorphisms within this gene have been associated with variation in RBC mean corpuscular volume and haemoglobin concentration in African-American populations[40]. Whether this reflects a direct influence of FAM234A on red blood cell characteristics or alternatively effects of a closely linked gene remains unclear.

In addition to evolutionary history and founder effects, the origin of increased variability in expression of FAM234A, SLC43A1 and SLC43A3 and other proteins in the Senegalese population may be explained by selective pressures from a variety of pathogens. These include (1) negative selective pressure against certain RBC surface receptors for *P. falciparum*, an endemic pathogen in sub-Saharan Africa; (2) previous negative selective pressure from *P. vivax*, which was but is no longer an endemic pathogen[41]; (3) current selective pressure by another pathogen (for example, arenaviruses such as Machupo virus). Nutritional differences between and within the studied populations may also modulate surface protein expression[42]. Future work will be required to address these questions in regards to data from the present study.

Overall, our data provides both a shortlist of potential novel receptors for *Plasmodium* for further study, in addition to comprehensive definition of the RBC surface proteome, which will facilitate diverse future studies of RBC biology.

## Methods

**RBC isolation.** Collection of clinical samples from healthy donors from the Senegalese National Blood Transfusion Center in Dakar, Senegal, and their experimental use were approved by the Ethics Committee of the Ministry of Health in Senegal and by the Institutional Review Board of the Harvard T.H. Chan School of Public Health. For samples derived from donors local to Cambridge, UK, ethical approval was obtained from University of Cambridge Human Biology Research Ethics Committee (HBREC.2016.011). In all, 2–3 ml of whole anticoagulated blood was collected from each donor with informed consent in accordance with the Declaration of Helsinki.

**Methods for RBC enrichment.** The following techniques were used alone or in combination to optimise enrichment of erythrocytes for plasma membrane (PM) proteomic analysis, as detailed in the Results section:

*Microcentrifugation*: 1 ml of whole blood was centrifuged at 1000 g for 5 min in a 1.5 ml Eppendorf tube. A 200 μl pipette tip was used to withdraw packed erythrocytes from the very bottom of the tube.

*Plasmodipur filtration to remove leukocytes*: Erythrocytes were resuspended in 50 ml room temperature PBS (Sigma). A Plasmodipur filter (EuroProxima) was mounted on a 50 ml syringe and gentle pressure was applied to filter cells.

*PBS washes to remove platelets and serum contaminants*: Erythrocytes were resuspended in 50 ml room temperature PBS (Sigma) and centrifuged at 400 g for 5 min. Residual PBS was aspirated.

*MACS microbead kits*: A CD45 microbead kit (Miltenyi Biotech, 135-045-801) was used with 250 μl whole blood to negatively select leukocytes. A CD235 microbead kit (Miltenyi Biotech, 130-050-501) was used with 100 μl whole blood to positively select erythrocytes. Both kits were used according to the manufacturer's instructions.

*Final optimised protocol (see also Results section)*: For each sample, 2x aliquots of 1 ml anticoagulated whole blood were centrifuged for 5 min at 1000 g in a 1.5 ml eppendorf. 150 μl was removed from the very bottom of each tube, and combined into 50 ml PBS, followed by Plasmodipur filtration. After centrifugation for 5 min

at 400 g, two further 50 ml PBS washes were performed. The resulting enriched RBC were directly subjected to PMP.

**RBC storage.** To facilitate simultaneous immunostaining for temporal flow cytometry analysis in supplementary fig. 6, RBC were stored in a high glycerol solution using a method adapted from a previously published protocol[43]. Briefly, RBC were isolated from packed spun whole blood as described above. Cells were then dispersed in a 40% glycerol solution and slowly cooled to −80 °C for storage. Glycerol was removed on the day of immunostaining and osmolality slowly reduced before returning to isotonic conditions. RBC were then immunostained as described below.

**Plasma membrane profiling.** We have described plasma membrane profiling previously[10,26], and details of the method used is recapitulated here with modifications for this study. For ~$1.5 \times 10^9$ of each enriched erythrocyte sample, surface sialic acid residues were oxidized with sodium meta-periodate (Thermo) then biotinylated with aminooxy-biotin (Biotium). After quenching, cells were incubated in 1% (v/v) Triton X-100 lysis buffer (10 mM Tris HCl, 1.6% Triton, 150 mM NaCl). Biotinylated glycoproteins were enriched with high affinity streptavidin agarose beads (Pierce) and washed extensively. Captured protein was denatured with dithiothreitol (DTT), alkylated with iodoacetamide (Sigma) and digested on-bead with trypsin (Promega) in 200 mM HEPES pH 8.5 for 3 h. Tryptic peptides were collected and labelled using TMT reagents. The reaction was quenched with hydroxylamine, and TMT-labelled samples combined in a 1:1:1:1:1:1:1:1:1:1 ratio. Labelled peptides were subjected to C18 solid-phase extraction (Sep-Pak, Waters) and vacuum-centrifuged to near-dryness.

**Offline fractionation and mass spectrometry.** The mass spectrometry protocols employed were similar to previous studies[44], and are recapitulated here with modifications for the present study. Mass spectrometry data was acquired using an Orbitrap Lumos for samples from UK donors, and an Orbitrap Fusion for samples from Senegal donors. In both cases, an Ultimate 3000 RSLC nano UHPLC equipped with a 300 μm ID × 5 mm Acclaim PepMap μ-Precolumn (Thermo Fisher Scientific) and a 75 μm ID × 50 cm 2.1 μm particle Acclaim PepMap RSLC analytical column was used.

*For Orbitrap Lumos experiments*: An unfractionated singleshot of 25% of the whole sample was analysed. Subsequently, the remaining 75% of the sample was separated into six fractions using tip-based strong cation exchange[26]. Loading solvent was 0.1% Formic Acid (FA) analytical solvent A: 0.1% FA and B: 80% acetonitrile + 0.1% FA. All separations were carried out at 55 °C. Samples were loaded at 5 μL/min for 5 min in loading solvent before beginning the analytical gradient. The singleshot sample was analysed on the following gradient: 3–7% B over 3 min, 7–37% B over 173 min, followed by a 4 min wash at 95% B and equilibration at 3% B for 15 min. Each of six fractions were analysed on the following gradient: 3–8% B over 3 min, 8–37% B over 88 min, followed by a 2 min wash at 95% B and equilibration at 3% B for 15 min. Each analysis used a MultiNotch MS3-based TMT method[45,46]. The following settings were used: MS1: 380–1500 Th, 120,000 Resolution, $2 \times 10^5$ automatic gain control (AGC) target, 50 ms maximum injection time. MS2: Quadrupole isolation at an isolation width of m/z 0.7, CID fragmentation (normalised collision energy (NCE) 35) with ion trap scanning in turbo mode from m/z 120, $1.5 \times 10^4$ AGC target, 120 ms maximum injection time. MS3: In Synchronous Precursor Selection mode the top three MS2 ions were selected for HCD fragmentation (NCE 65) and scanned in the Orbitrap at 60,000 resolution with an AGC target of $1 \times 10^5$ and a maximum accumulation time of 150 ms. Ions were not accumulated for all parallelisable time. The entire MS/MS/MS cycle had a target time of 3 s. Dynamic exclusion was set to ±10 ppm for 70 s. MS2 fragmentation was triggered on precursors $5 \times 10^3$ counts and above.

*For Orbitrap Fusion experiments*: As the samples analysed on the Orbitrap Lumos contained substantial peaks likely to relate to abundant contaminants such as trypsin, TMT-labelled tryptic peptides were instead separated by high pH reversed-phase (HpRP) fractionation[44]. Peptide containing fractions were then orthogonally recombined into six fractions. Fractions identified by UV absorbance as containing very abundant peptide were omitted from this recombination procedure and instead combined into a separate, seventh fraction. For LC/MS3, loading solvent was 0.1% trifluoroacetic acid (TFA), analytical solvent A: 0.1% FA and B: acetonitrile + 0.1% FA. All separations were carried out at 55 °C. Samples were loaded at 10 μl/min for 5 min in loading solvent before beginning the analytical gradient. The following gradient was used: 3–8% B over 3 min, 8–37% B over 85 min, 37–95% B over 4 min followed by a 3 min wash at 95% B and equilibration at 3% B for 15 min. Each analysis used a MultiNotch MS3-based TMT method[45,46]. The following settings were used: MS1: 400–1400 Th, Quadrupole isolation, 120,000 Resolution, $2 \times 10^5$ AGC target, 50 ms maximum injection time, ions injected for all parallisable time. MS2: Quadrupole isolation at an isolation width of m/z 0.7, CID fragmentation (NCE 30) with ion trap scanning out in rapid mode from m/z 120, $1 \times 10^4$ AGC target, 70 ms maximum injection time, ions accumulated for all parallisable time in centroid mode. MS3: in Synchronous Precursor Selection mode the top 10 MS2 ions were selected for HCD fragmentation (NCE 65) and scanned in the Orbitrap at 50,000 resolution with an AGC target of $5 \times 10^4$ and a maximum accumulation time of 150 ms, ions were not

accumulated for all parallelisable time. The entire MS/MS/MS cycle had a target time of 3 s. Dynamic exclusion was set to ±10 ppm for 90 s. MS2 fragmentation was trigged on precursors $5 \times 10^3$ counts and above.

**Quantification and statistical analysis.** *Data analysis*: the data analysis protocols employed were similar to previous studies[44], and are recapitulated here with modifications for the present study. In the following description, we list the first report in the literature for each relevant algorithm. Mass spectra were processed using a Sequest-based software pipeline for quantitative proteomics, 'MassPike', through a collaborative arrangement with Professor Steve Gygi's laboratory at Harvard Medical School. MS spectra were converted to mzXML using an extractor built upon Thermo Fisher's RAW File Reader library (version 4.0.26). In this extractor, the standard mzXML format has been augmented with additional custom fields that are specific to ion trap and Orbitrap mass spectrometry and essential for TMT quantitation. These additional fields include ion injection times for each scan, Fourier Transform-derived baseline and noise values calculated for every Orbitrap scan, isolation widths for each scan type, scan event numbers, and elapsed scan times. This software is a component of the MassPike software platform and is licensed by Harvard Medical School.

A combined database was constructed from the human Uniprot database (26 January 2017), and common contaminants such as porcine trypsin. The combined database was concatenated with a reverse database composed of all protein sequences in reversed order. Searches were performed using a 20 ppm precursor ion tolerance[47]. Product ion tolerance was set to 0.03 Th. TMT tags on lysine residues and peptide N termini (229.162932 Da) and carbamidomethylation of cysteine residues (57.02146 Da) were set as static modifications, while oxidation of methionine residues (15.99492 Da) was set as a variable modification.

To control the fraction of erroneous protein identifications, a target-decoy strategy was employed[48,49]. Peptide spectral matches (PSMs) were filtered to an initial peptide-level false discovery rate (FDR) of 1% with subsequent filtering to attain a final protein-level FDR of 1%[50,51]. PSM filtering was performed using a linear discriminant analysis[52]. This distinguishes correct from incorrect peptide IDs in a manner analogous to the widely used Percolator algorithm[53], though employing a distinct machine learning algorithm. The following parameters were considered: XCorr, ΔCn, missed cleavages, peptide length, charge state, and precursor mass accuracy. Protein assembly was guided by principles of parsimony to produce the smallest set of proteins necessary to account for all observed peptides[52].

Proteins were quantified by summing TMT reporter ion counts across all matching peptide-spectral matches using 'MassPike'[45,46]. Briefly, a 0.003 Th window around the theoretical m/z of each reporter ion (126, 127n, 127c, 128n, 128c, 129n, 129c, 130n, 130c, 131n) was scanned for ions, and the maximum intensity nearest to the theoretical m/z was used. The primary determinant of quantitation quality is the number of TMT reporter ions detected in each MS3 spectrum, which is directly proportional to the signal-to-noise (S:N) ratio observed for each ion[54]. Conservatively, every individual peptide used for quantitation was required to contribute sufficient TMT reporter ions (minimum of ~1250 per spectrum) so that each on its own could be expected to provide a representative picture of relative protein abundance[45]. An isolation specificity filter was additionally employed to minimise peptide co-isolation[55]. Peptide-spectral matches with poor quality MS3 spectra (more than 9 TMT channels missing and/or a combined S:N ratio of <250 across all TMT reporter ions) or no MS3 spectra at all were excluded from quantitation. Peptides meeting the stated criteria for reliable quantitation were then summed by parent protein, in effect weighting the contributions of individual peptides to the total protein signal based on their individual TMT reporter ion yields. Protein quantitation values were exported for further analysis in Microsoft Excel.

For protein quantitation, reverse and contaminant proteins were removed. Despite extensive washing of biotinylated proteins when bound to Streptavidin beads, variable levels of contamination with abundant haemoglobin components were nevertheless detectable. As opposed to normalisation assuming equal protein loading across all channels, normalisation was instead performed from the summed signal:noise values of the three most abundant RBC surface proteins (which are also all recognised cell surface blood group proteins), GlycophorinA (GYPA), Glycophorin C (GYPC) and SLC4A1 (see Results). For further analysis and display in figures, fractional TMT signals were used (i.e. reporting the fraction of maximal signal observed for each protein in each TMT channel, rather than the absolute normalized signal intensity). This effectively corrected for differences in the numbers of peptides observed per protein.

Proteins were filtered to include those most likely to be present at the cell surface with high confidence. These comprised proteins with Uniprot Subcellular Location terms matching 'Multipass', 'GPI anchored', 'Lipid Anchored', 'Type I transmembrane', 'Type II transmembrane', 'Type III transmembrane', 'Type IV transmembrane', and those predicted to have transmembrane regions based on TMHMM version 2.0[29]. To be included in the analysis, a minimum of two peptide quantifications were required per protein (which could be derived exclusively from either the UK or Senegal samples, or one peptide from both samples). Clusters C1 and C2 (Supplementary Fig. 1D) identified a list of proteins likely to reflect

abundant serum contaminants. Data were filtered to exclude these contaminants, in total removing six proteins (IGHM, FGG, C9, C8B, CD14 and ITIH1). Three HLA-A species were identified in our data, likely derived from low-level contamination with leukocytes (<0.05%, Supplementary Fig. 1B). To enable comparison with previous literature, signal:noise values for all three alleles were summed to give a single combined result for HLA-A.

For Supplementary Fig. 1D, the Database for Annotation, Visualisation and Integrated Discovery (DAVID) was used to determine pathway enrichment[56]. Clusters C1 and C2 were searched against a background of all proteins quantified.

Hierarchical centroid clustering based on Euclidian distance or uncentered correlation was performed using Cluster 3.0 (Stanford University) and visualised using Java Treeview (http://jtreeview.sourceforge.net). Subcellular Location terms were added from www.uniprot.org.

*iBAQ*: To determine protein abundance, RAW files for a given donor set (UK or Senegal) were re-analysed using MaxQuant 1.6.0.13 using the same human Uniprot database as described above. iBAQ values were calculated and exported to Microsoft Excel for further analysis. In the UK experiment, a single donation was included that failed for technical reasons. For each protein, the iBAQ values were adjusted in proportion to the total contribution derived from that donor, calculated from the relative TMT abundance determined in MaxQuant.

iBAQ is based on the summed intensity of all peptides from each protein divided by the number of observable peptides. Therefore the iBAQ abundance is composed of three parts, which contribute to the overall dynamic range of measurement. These are (a) MS1 precursor intensity for each peptide; (b) number of peptides observed for each protein and (c) number of observable peptides from each protein. For example, amongst proteins quantified from Senegalese donors, ADAM9 had one of the lowest iBAQ abundances, of $3.8 \times 10^5$, and was quantified by three peptides. By comparison, GYPC (Glycophorin C) had the highest abundance ($9.1 \times 10^{10}$), and was quantified by 154 peptides. ADAM9 is composed of 819 amino acids (41 theoretically observable peptides), whereas GYPC is 128 amino acids long (two theoretically observable peptides). The iBAQ abundance of GYPC is higher for three reasons; (a) higher summed MS1 precursor intensities; (b) greater number of observed peptides; (c) lower number of observable peptides.

Estimates of protein copy number per red blood cell were derived from the fraction each protein contributed to the total iBAQ signal multiplied by an average estimated red blood cell surface protein copy number ($1.74 \times 10^7$) derived from previous studies[18,25].

*Literature comparison*: the existing literature on whole red blood cell proteomes was reviewed, and the 10 highest quality mass spectrometry analyses from 2006 onwards were included in our comparison, starting with Pasini et al.[16,17,19–25]. Prior to this point, most analyses had been based on 2D gel electrophoresis, identifying a very limited range of proteins, Proteins from each manuscript were extracted from published data resources. For data that included protein UniProt identifiers, no further modification was required. For other data, the following protein identification conversion tools were used: UniProt (https://www.uniprot.org/uniprot/), DAVID (https://david.ncifcrf.gov/) and BioDB (https://biodbnet-abcc.ncifcrf.gov/db/db2db.php)[56–59].

*Flow cytometric analysis of RBC surface proteins*: $2 \times 10^7$ enriched RBC were incubated with Fc blocking reagent then specific fluorophore-conjugated antibodies at 4 °C for 30 min with Biolegend Human Trustain FcX. Antibodies used were: anti-CD45 (Becton-Dickinson, 555482); anti-CD61 (Becton-Dickinson, 555754), anti-ICAM4 (R&D Biosystems, FAB8397A), anti-CD35 (Miltenyi Biotec, 130-099-913), anti-GYPA (Miltenyi Biotec, 130-100-269), anti-SLC4A1 (Miltenyi Biotec, 130-105-728). Cells were washed in PBS/0.4% (v/v) citrate/0.5% (v/v) BSA prior to fixation and analysis on a BD LSR Fortessa. The only major deviation from this protocol was for staining for SLC4A1, for which the antibody recognises an intracellular epitope. As such RBC were treated with Fc blocking reagent for 30 min prior to being permeabilise using a Cytofix/Cytoperm kit (Becton Dickinson), then immunostained prior to flow cytometric analysis. After gating by forward and side-scatter, ≥20,000 events were captured. Data was analysed using FlowJo V10 (FlowJo, LLC). Data for each maker was acquired from an independently stained sample.

*Statistics and reproducibility*: For Fig. 2a, the method of significance A was used to estimate the p-value that each protein ratio (iBAQ Senegal: iBAQ UK) was significantly different to 1[60]. As an increase in variation in iBAQ values was seen as expected as overall protein abundance decreased, for the purposes of significance A calculation, ratios were split into two 'bins' at the mid-point of abundance in the Senegalese samples ($\log_{10}$ 8.5). Values were calculated and corrected for multiple hypothesis testing using the method of Benjamini-Hochberg in Perseus version 1.5.1.6[60]. A corrected *p*-value of <0.01 (two-tailed) was considered statistically significant. Calculations were based on proteins identified with both %CV and iBAQ values in our 'sensitive' protein list with contaminants and duplicates removed ($n = 165$).

For comparing the distribution of %CV values between proteins identified in UK and Senegalese donors a two-tailed Mann–Whitney U-test was performed on proteins found in both populations using XLStat (Addinsoft). *P*-values were calculated using the Monte Carlo method with 100,000 iterations.

For flow cytometry validation of proteomic data, small volumes of blood were taken from eight donors, enriched for RBC and split into samples stained independently for each marker. A single sample was discarded due to poor sample quality.

**Reporting summary**. Further information on research design is available in the Nature Research Reporting Summary linked to this article.

## Data availability

The mass spectrometry proteomics data has been deposited to the ProteomeXchange Consortium (http://www.proteomexchange.org/) via the PRIDE partner repository. Project accession number PXD013852, project name 'Quantitative comparative analysis of human erythrocyte surface proteins between individuals from two genetically distinct populations'. All raw flow cytometry data will be made available upon request.

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

## Acknowledgements

We are grateful to Prof. Steven Gygi for providing access to the 'MassPike' software pipeline for quantitative proteomics. This work was supported by a Wellcome Trust Senior Clinical Research Fellowship (108070/Z/15/Z) to M.P.W., a strategic award to Cambridge Institute for Medical Research from the Wellcome Trust (100140), NIH Grants R01AI091787 and R01HL139337 and Bill and Melinda Gates Foundation Grant OPP1023594 to M.T.D. B.J.R. was supported by an Evelyn Trust Fellowship (18/27). U.K. was supported by a Canadian Institutes of Health Research Postdoctoral Fellowship. This study was additionally supported by the Cambridge Biomedical Research Centre, UK.

## Author contributions

B.J.R., U.K., M.T.D., M.P.W. designed the experiments. B.J.R., U.K., A.A., L.N., J.W., R.A., M.T.D., M.P.W. performed the experiments. B.J.R., J.M.G., M.P.W. analysed the proteomics data. B.J.R., M.P.W. wrote the paper. B.J.R., U.K., A.A., T.D., M.T.D., M.P.W. edited the paper. M.P.W. and M.T.D. supervised all research.

## Additional information

**Competing interests:** The authors declare no competing interests.

