## [Peer Review File · Communications Biology]

Reviewers' comments:

Reviewer #1 (Remarks to the Author):

In this work, the authors employed plasma membrane profiling and tandem mass tag-based mass spectrometry to specifically quantify cell surface proteins from primary red blood cells (RBC). They analyzed RBCs from two sets of nine donors from the UK and Senegal. In combination with protein annotations, they found the 'stringent' and 'sensitive' RBC surface proteomes and identified candidate Plasmodium receptors on the basis either of diminished protein abundance, or increased variation in West African individuals compared to people from the UK. The method has been reported previously, as cited in this manuscript (Ref. 6 and 27). It is interesting to analyze the surface proteomes in RBC, and this work also provides useful information.

Because of the contamination, they filtered the data set based on protein annotations or structure information, as described on Page 8: "we took a stringent approach requiring 2 peptides for each identified protein and at least one of the following annotations derived from the Uniprot subcellular localisation database: 'Multipass', 'GPI anchored', 'Lipid Anchored', 'Type I transmembrane', 'Type II transmembrane', 'Type III transmembrane', 'Type IV transmembrane', or ≥ 1 predicted transmembrane regions based on TMHMM version 2.0." It might be better to provide more detailed information about protein identifications. For example, totally how many proteins were identified in their experiments? How many contain two peptides (unique or total)?

Minor point:

Figures need to be better organized. There are much empty space in Figures 1 and 4.

Reviewer #2 (Remarks to the Author):

This paper aims to improve on the current understanding of erythrocyte surface proteins by employing an advanced proteomic approach that allows improved purification as well as quantification. Comparison of the data set obtained here with previously published studies provides a strong basis for determining the most comprehensive red blood cell surface proteome to date. This combined data set than was utilized to study the variation of the proteome in different populations. This allowed the identification of a number of protein that are under apparent selection in the different populations possibly providing an opportunity to link receptors to specific diseases.

The data generated is clearly of interest to a broad research community and while the methods used are not themselves novel their application to study the rbc surface proteome is a new application. Overall the study is well presented and sound and it will be clearly valuable to the community. At the same time there are some issues that I feel should be addressed before publication.

1. The authors state that there is an approximately 10^5 fold difference between the maximum and minimum number of proteins expressed. I am wondering whether the authors can clarify whether these numbers fall within the dynamic range of detection by the approach used.
2. On the same note these quantitative differences are significant and I am wondering whether the authors could address the question on how many molecules per cell this would relate to and whether some of the differences represent different sub populations of rbc with some expressing the protein and others not.
3. Can the authors establish whether the overall quantity of protein is relatively stable in an individual or there are dynamic variations that happen over time? IT would for example be interesting to know if using the FACS approach used in this study similar results in terms of abundance are obtained over a period of time from the same individual

4. Figure 4A it should be mentioned in the legend what is in x – Axis and what is in y-axis. For 4B the different spots should be identified in terms of the protein they refer to.

Reviewer #3 (Remarks to the Author):

In their manuscript “Quantitative comparative analysis of human erythrocyte surface proteins between individuals from two genetically distinct populations”, Ravenhill et al. use a novel proteomic strategy to selectively enrich and characterize proteins exposed at the surface of the cell membrane of red blood cells in humans. This interesting proteomic method is used to map variances between individuals from specific genetic populations, exposed to different selection pressure by presence of different medical relevant parasites (*Plasmodium* species) in their environment. This allows the authors to identify potential new receptors for the entry of *Plasmodium* into the red blood cells.

In general, the manuscript is well written and technological developments are nicely combined with a medical and biological highly relevant question. Most often, the biological relevant output from spatial proteomic studies is limited, as just one “enriched” fraction of an organelle or membrane is analyzed, thereby it is often not possible to determine bone-fide compartment proteins from contaminants, especially with increased sensitivity of mass spectrometry. This is especially relevant for the red blood cells, that are impossible to purify to complete homogeneity due to presence of other cell types and especially contaminations from highly abundant blood proteins. Here, an initial profiling step is performed to identify contaminants. In a second step, cell surface proteins are biotinylated and enriched. This complex workflow strongly increases the reliability of the identified cell surface proteome. The quality of the proteomic analysis of this study is high. However, no follow up experiments were performed to characterize the functions of the identified red blood cell receptors that vary in the surface levels.

In this study, the data analysis is already well elaborated. For example, normalization problems that varying amounts of plasma contamination in the different would impose are already addressed by an alternative normalization method that is based on levels of three highly abundant receptors with only little variations in their expression levels (Figure S4).

The manuscript could be accepted for publication if the following points are addressed:

- Figure S1: An overview figure should be added how the red blood cells were enriched and purified, including the TMT labeling strategy. In Figure S1d, the figure would be easier to interpret for the reader if three colors (for example blue, white, red) would be used instead of the red-black gradient.
- Figure 1B: In the text, details should be given of how many proteins were quantified in total and how many of those proteins are then discarded in each of the filtering steps.
- Limitations of the study should be more clearly stated:
 - o Number of individuals included in this study is low, limiting the statistical relevance
 - o It should be clearly stated in the abstract that “potential” new receptors are identified, as any biological follow up that proves this is missing. This point should also be given more weight in the discussion.

Point-by-point response to the referees' comments:

Reviewer #1

In this work, the authors employed plasma membrane profiling and tandem mass tag-based mass spectrometry to specifically quantify cell surface proteins from primary red blood cells (RBC). They analyzed RBCs from two sets of nine donors from the UK and Senegal. In combination with protein annotations, they found the 'stringent' and 'sensitive' RBC surface proteomes and identified candidate Plasmodium receptors on the basis either of diminished protein abundance, or increased variation in West African individuals compared to people from the UK. The method has been reported previously, as cited in this manuscript (Ref. 6 and 27). It is interesting to analyze the surface proteomes in RBC, and this work also provides useful information.

Because of the contamination, they filtered the data set based on protein annotations or structure information, as described on Page 8: "we took a stringent approach requiring 2 peptides for each identified protein and at least one of the following annotations derived from the Uniprot subcellular localisation database: 'Multipass', 'GPI anchored', 'Lipid Anchored', 'Type I transmembrane', 'Type II transmembrane', 'Type III transmembrane', 'Type IV transmembrane', or ≥ 1 predicted transmembrane regions based on TMHMM version 2.0."

It might be better to provide more detailed information about protein identifications. For example, totally how many proteins were identified in their experiments? How many contain two peptides (unique or total)?

We thank the reviewer for their complimentary assessment of our work. As suggested, we have now added detail to the text of the total number of proteins identified and discarded in each filtering step. We additionally include the number of single and ≥ 2 peptide-containing proteins.

Minor point:

Figures need to be better organized. There are much empty space in Figures 1 and 4.

Many thanks for this suggestion. We have better organised Figures 1 and 4 as suggested.

Reviewer #2 (Remarks to the Author):

This paper aims to improve on the current understanding of erythrocyte surface proteins by employing an advanced proteomic approach that allows improved purification as well as quantification. Comparison of the data set obtained here with previously published studies provides a strong basis for determining the most comprehensive red blood cell surface proteome to date. This combined data set than was utilized to study the variation of the proteome in different populations. This allowed the identification of a number of protein that are under apparent selection in the different populations possibly providing an opportunity to link receptors to specific diseases.

The data generated is clearly of interest to a broad research community and while the methods used are not themselves novel their application to study the rbc surface proteome is a new application.

Overall the study is well presented and sound and it will be clearly valuable to the community. At the same time there are some issues that I feel should be addressed before publication.

We thank the reviewer for their complimentary assessment of our work.

1. *The authors state that there is an approximately 10^5 fold difference between the maximum and minimum number of proteins expressed. I am wondering whether the authors can clarify whether these numbers fall within the dynamic range of detection by the approach used.*

We found that the most highly expressed proteins were present at $\sim 10^5$ greater abundance than the least expressed proteins. These estimates were generated using identity based absolute quantitation (iBAQ) to compare the abundance of each protein in our 'sensitive' list and thus contribution of each protein to the RBC cell surface proteome. iBAQ is based on the summed intensity of all peptides from each protein divided by the number of observable peptides. Therefore the iBAQ abundance is composed of three parts, which contribute to the overall dynamic range of measurement. These are (a) MS1 precursor intensity for each peptide; (b) number of peptides observed for each protein and (c) number of observable peptides from each protein. For example, amongst proteins quantified from Senegalese donors, ADAM9 had one of the lowest iBAQ abundances, of 3.8×10^5 , and was quantified by three peptides. By comparison, GYPC (Glycophorin C) had the highest abundance (9.1×10^{10}), and was quantified by 154 peptides. ADAM9 is composed of 819 amino

acids (41 theoretically observable peptides), whereas GYPC is 128 amino acids long (2 theoretically observable peptides). The iBAQ abundance of GYPC is higher for three reasons; (a) higher summed MS1 precursor intensities; (b) greater number of observed peptides; (c) lower number of observable peptides.

This dynamic range corresponds to previous data; for example, in the original description of the iBAQ algorithm, Schwanhäusser et al found that iBAQ abundances of proteins in NIH3T3 fibroblasts spanned more than five orders of magnitude. These measurements were validated by independent spike-in experiments using a mixture of 48 proteins with known concentrations. iBAQ values correlated well with known absolute protein amounts over at least four orders of magnitude and had a higher precision and accuracy than alternative measures of absolute protein abundance (Schwanhausser et al, Nature 2011).

We have added the above details to the results section and supplemental methods.

2. *On the same note these quantitative differences are significant and I am wondering whether the authors could address the question on how many molecules per cell this would relate to and whether some of the differences represent different sub populations of rbc with some expressing the protein and others not.*

To translate iBAQ abundances into protein copy number, it is essential to know the total amount of biotinylated plasma membrane protein per sample. However, this is necessarily small and difficult to measure precisely, in our experience at best giving values at the very lowest end of commercially available assays.

An alternative approach to estimate protein copy number is to adjust our measured iBAQ values by the total number of RBC membrane proteins. Two previous studies gave concordant values, derived from white ghost membranes: 1.68×10^7 molecules per membrane, and 1.8×10^7 (range $1.5 - 2.4 \times 10^7$) molecules per membrane, giving an average $\sim 1.74 \times 10^7$ molecules per membrane (Gautier et al Cell Reports 2016, Bryk et al J Prot Res 2017). We have therefore estimated the number of cell surface molecules per RBC for each protein by (protein iBAQ abundance / summed iBAQ abundance for all proteins) * 1.74×10^7 , and have added these values to the relevant supplemental tables. For each protein, the estimates for number of molecules per cell range from 6 - $\sim 4,500,000$ per RBC cell surface.

It is unlikely that the differences in protein abundance we observed are due to different RBC subpopulations as each flow cytometry-based analysis did not observe any bimodal or higher order patterns of staining. Similarly, the majority of published healthy RBC flow cytometry shows similar unimodal distributions (for example Brown et al Clinical Chemistry 2000, Magor et al Blood 2015, Zenonos et al JEM 2015).

3. *Can the authors establish whether the overall quantity of protein is relatively stable in an individual or there are dynamic variations that happen over time? IT would for example be interesting to know if using the FACS approach used in this study similar results in terms of abundance are obtained over a period of time from the same individual*

Many thanks for this suggestion. To address this point, we harvested RBC from two donors at 2-weekly intervals, then analysed the RBC by flow cytometry for GYPA, ICAM4 and CR1 expression (data below). Results suggest that within a single donor the surface levels of these proteins are stable over time, with %CV across the time points of <20%.

Figure S6

4. Figure 4A it should be mentioned in the legend what is in x – Axis and what is in y-axis. For 4B the different spots should be identified in terms of the protein they refer to.

Many thanks for this suggestion; we have now updated the Figure 4A legend and Figure 4B diagram as suggested.

Reviewer #3 (Remarks to the Author):

In their manuscript “Quantitative comparative analysis of human erythrocyte surface proteins between individuals from two genetically distinct populations”, Ravenhill et al. use a novel proteomic strategy to selectively enrich and characterize proteins exposed at the surface of the cell membrane of red blood cells in humans. This interesting proteomic method is used to map variances between individuals from specific genetic populations, exposed to different selection pressure by presence of different medical relevant parasites (*Plasmodium* species) in their environment. This allows the authors to identify potential new receptors for the entry of *Plasmodium* into the red blood cells.

In general, the manuscript is well written and technological developments are nicely combined with a medical and biological highly relevant question. Most often, the biological relevant output from spatial proteomic studies is limited, as just one “enriched” fraction of an organelle or membrane is analyzed, thereby it is often not possible to determine bone-fide compartment proteins from contaminants, especially with increased sensitivity of mass spectrometry. This is especially relevant for the red blood cells, that are impossible to purify to complete homogeneity due to presence of other cell types and especially contaminations from highly abundant blood proteins. Here, an initial profiling step is performed to identify contaminants. In a second step, cell surface proteins are biotinylated and enriched. This complex workflow strongly increases the reliability of the identified cell surface proteome. The quality of the proteomic analysis of this study is high. However,

no follow up experiments were performed to characterize the functions of the identified red blood cell receptors that vary in the surface levels.

In this study, the data analysis is already well elaborated. For example, normalization problems that varying amounts of plasma contamination in the different would impose are already addressed by an alternative normalization method that is based on levels of three highly abundant receptors with only little variations in their expression levels (Figure S4).

We thank the reviewer for their complimentary assessment of our work.

The manuscript could be accepted for publication if the following points are addressed:

- *Figure S1: An overview figure should be added how the red blood cells were enriched and purified, including the TMT labeling strategy. In Figure S1d, the figure would be easier to interpret for the reader if three colors (for example blue, white, red) would be used instead of the red-black gradient.*

Many thanks for this suggestion. We have now added an overview figure to Figure S1, and modified Figure S1D to employ three colours.

- *Figure 1B: In the text, details should be given of how many proteins were quantified in total and how many of those proteins are then discarded in each of the filtering steps.*

As also suggested by Reviewer #1, we have now included these details in the text.

- *Limitations of the study should be more clearly stated:*

o *Number of individuals included in this study is low, limiting the statistical relevance*

o *It should be clearly stated in the abstract that “potential” new receptors are identified, as any biological follow up that proves this is missing. This point should also be given more weight in the discussion.*

We have now considered these points in the discussion, and have added details to the abstract that “potential” new receptors are identified as suggested.

REVIEWERS' COMMENTS:

Reviewer #2 (Remarks to the Author):

I feel the authors have addressed the outstanding issues satisfactorily.

Reviewer #3 (Remarks to the Author):

The authors have addressed all points and suggestions in their revisions.